# DisProtBench: A Disorder-Aware, Task-Rich Benchmark for Evaluating Protein Structure Prediction in Realistic Biological Contexts

## Abstract

Recent advances in protein structure prediction have achieved near-atomic accuracy for well-folded proteins. However, current benchmarks inadequately assess model performance in biologically challenging contexts, especially those involving intrinsically disordered regions (IDRs), limiting their utility in applications like drug discovery, disease variant interpretation, and protein interface design. We introduce DisProtBench, a comprehensive benchmark for evaluating protein structure prediction models (PSPMs) under structural disorder and complex biological conditions. DisProtBench spans three key axes: (1) Data complexity—covering disordered regions, G protein-coupled receptors (GPCR)–ligand pairs, and multimeric complexes; (2) Task diversity—benchmarking eleven leading PSPMs across structure-based tasks with unified classification, regression, and interface metrics; and (3) Interpretability—via the DisProtBench Portal, offering precomputed 3D structures and visual error analyses. Our results reveal significant variability in model robustness under disorder, with low-confidence regions linked to functional prediction failures. Notably, global accuracy metrics often fail to predict task performance in disordered settings, emphasizing the need for function-aware evaluation. DisProtBench establishes a reproducible, extensible, and biologically grounded framework for assessing next-generation PSPMs in realistic biomedical scenarios.

## 1 Introduction

Many critical cellular processes—such as signal transduction, transcriptional regulation, and molecular recognition—are mediated not by static, well-folded domains, but by intrinsically disordered regions (IDRs) Bondos et al. (2022); Trivedi & Nagarajaram (2022); Madhurima et al. (2023). These regions lack stable conformations, exhibit high conformational variability, and frequently engage in context-dependent interactions Wright & Dyson (2015); Uversky (2019), making them indispensable for biological function but elusive to both experimental characterization and computational modeling. Given their centrality in modulating protein–protein interactions and dynamic assemblies, understanding IDRs is a core challenge in molecular biology. In response to this challenge, the field has seen rapid progress in deep learning-based models for predicting protein structure and interactions. Models such as AlphaFold (AF) Jumper et al. (2021); Hu & Ohue (2024a), and ESMFold Lin et al. (2023) have achieved impressive accuracy in predicting the structures of well-ordered protein domains and their binary or multimeric interactions. These advances have had a significant downstream impact, improving drug discovery pipelines Chang et al. (2024), accelerating protein function annotation Anfinsen (1973); Dill et al. (2008), and elucidating the structural basis of disease Abramson et al. (2024a).

However, these models largely overlook the disordered regions that dominate interaction networks in higher organisms. While benchmarks such as Critical Assessment of Structure Prediction (CASP) Moult et al. (2018) assess global folding accuracy and Critical Assessment of Intrinsic protein Disorder (CAID) Necci et al. (2021); Conte et al. (2023) addresses binary IDR classification, there is no unified evaluation framework that captures the full spectrum of structural uncertainty, functional relevance, and context-specific behavior associated with IDRs. In particular, existing benchmarks fall short in assessing model performance on disordered regions involved in protein–protein and protein–ligand interactions, or in evaluating their impact on real-world biological tasks. To fill this

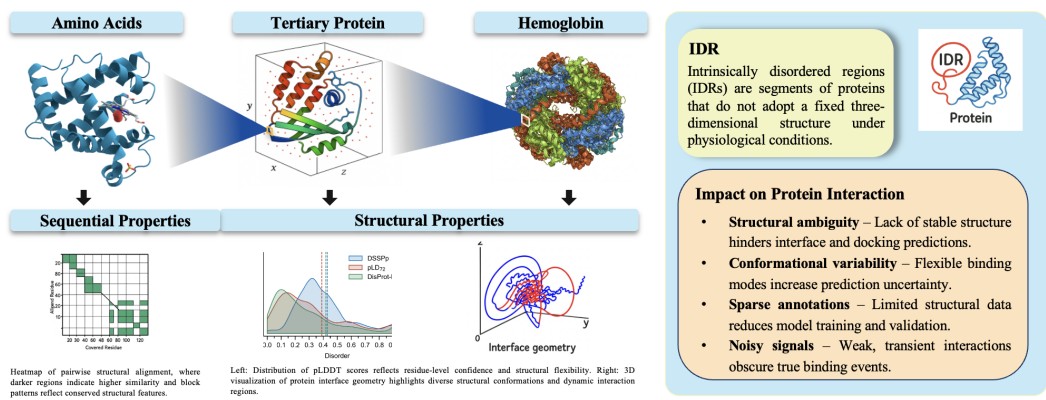

Figure 1: Overview of IDRs and their impact on protein interaction prediction.

critical gap, we introduce **DisProtBench**, a disorder-aware, task-rich benchmark designed to evaluate structure and interaction prediction models under biologically realistic and functionally complex conditions. DisProtBench captures diverse interaction modalities—spanning disordered regions, multimeric complexes, and ligand-bound conformations—enabling more meaningful assessments of model robustness, failure modes, and translational utility. DisProtBench makes the following key contributions:

**1. Database Development:** We curate a diverse benchmark dataset covering multiple biologically complex scenarios involving IDRs. This includes thousands of disease-associated human proteins with disordered regions, G protein-coupled receptors (GPCR)–ligand interactions relevant to drug discovery, and multimeric complexes with disorder-mediated interfaces. The dataset captures structural heterogeneity critical for evaluating model robustness in realistic biological contexts.

**2. Task and Toolbox Development:** We design an extensible evaluation toolbox to benchmark eleven state-of-the-art PSPMs across disorder-sensitive tasks, including protein–protein interaction (PPI) prediction, ligand-binding affinity estimation, and inter-residue contact mapping. The toolbox supports unified classification, regression, and interface metrics, enabling systematic evaluation of functional reliability in domains such as drug discovery and protein engineering. We also adopt predicted Local Distance Difference Test (pLDDT)-based stratification throughout our evaluation, which allows us to isolate model behavior in ambiguous regions. To our knowledge, DisProtBench is the first benchmark to formalize and operationalize this strategy across multiple tasks and model families.

**3. Visual-Interactive Interface Development:** The DisProtBench Portal provides precomputed 3D structure visualizations, cross-model comparison heatmaps, and interactive panels for downstream task results. It allows users to explore structure–function relationships, investigate disorder-specific performance variation, and interpret error patterns without requiring local model execution, facilitating hypothesis generation and human–AI collaboration.

DisProtBench bridges structural fidelity with biological relevance, enabling robust, functionally aware structure predictors and advancing our understanding of proteins at the order–disorder boundary. Code, data, and portal are available at DisProtBench.

## 2 RELATED WORK

**Homology Modeling and Community Benchmarks.** Early computational approaches, such as homology modeling and fragment-based assembly (e.g., Rosetta Rohl et al. (2004)), achieved partial success by leveraging structural databases. To systematize evaluation, the CASP initiative was launched in the 1990s Moult et al. (1995; 2018), providing blind prediction challenges and standardized metrics like Root Mean Square Deviation (RMSD) and Global Distance Test-Total Score (GDT-TS). However, CASP primarily focused on structured, single-chain proteins and did not explicitly account for biological complexity such as intrinsic disorder or multimeric interactions.

**Deep Learning Revolution.** The emergence of AF2 Jumper et al. (2021); Tunyasuvunakool et al. (2021) and RoseTTAFold Baek et al. (2021) marked a paradigm shift, achieving near-atomic accuracy for well-folded proteins. These models solved the core challenges of global fold prediction and significantly outperformed classical methods. Extensions such as ESMFold Lin et al. (2023) and AF3 Krokidis et al. (2025) expanded this capability using evolutionary-scale language models and integrated ligand/complex prediction, respectively. Yet, these systems remain limited in modeling flexible, functionally ambiguous structures such as IDRs Chakravarty et al. (2025); Wilson et al. (2022).

**Intrinsic Disorder and Functional Complexity.** IDRs are critical for signaling, regulation, and disease, often mediating transient or context-specific interactions Wright & Dyson (2015); Uversky (2019); Oldfield & Dunker (2014). Their conformational heterogeneity makes them difficult to predict Dill et al. (2008); Dunker et al. (2001). The CAID benchmark Necci et al. (2021); Conte et al. (2023) evaluates sequence-based disorder predictors using binary metrics like F1, Precision, and Area Under the Curve (AUC), but overlooks conformational variability, functional relevance, and downstream performance. Recent studies Lotthammer et al. (2024); Erdős & Dosztányi (2024) argue that these metrics are insufficient to capture the full spectrum of the disorder. While confidence scores such as pLDDT Jumper et al. (2021); Wilson et al. (2022) have been proposed as proxies for structural ambiguity, their connection to biological function remains underexplored. A unified framework linking structural uncertainty to real-world utility—such as performance in PPI or ligand binding—remains an open challenge.

**Protein–Protein and Protein–Ligand Interactions.** Structure-based prediction of PPIs and ligand-binding sites is central to many applications in drug discovery Vangone & Bonvin (2015); Dhakal et al. (2022). While progress has been made using machine learning and deep learning Zhang et al. (2024); Xia et al. (2024), these methods typically assume static and complete input structures. Disorder-mediated binding interfaces, especially in multimeric assemblies or GPCR-ligand systems, remain underexplored. Although AF-Multimer has enabled limited multichain modeling Evans et al. (2021); Hu & Ohue (2024a), there is no benchmark that systematically evaluates such cases under structural uncertainty.

**Interpretability and Usability in Model Evaluation.** A critical yet often overlooked dimension is the interpretability of model predictions and their usability by non-experts. While tools like PyMOL Schrödinger, LLC (2015) allow detailed inspection, few benchmarks provide precomputed visualizations or comparative error analysis Chang et al. (2024). This gap hampers practical deployment and limits scientific insight.

DisProtBench addresses these longstanding gaps by introducing a comprehensive, disorder-aware benchmark with thousands of proteins across three mainstream tasks and integrates three axes: (1) biologically grounded datasets encompassing IDRs, ligand-bound GPCRs, and multimeric complexes; (2) downstream tasks including PPI and drug binding with unified structural and functional metrics; and (3) an interactive portal enabling 3D visualization, model comparison, and error diagnosis without local computation. By aligning predictive evaluation with biological complexity and interpretability, DisProtBench advances the field toward more realistic, functional, and accessible protein modeling.

## 3 OVERVIEW OF DISPROTBENCH

Most state-of-the-art protein structure prediction frameworks follow a common pipeline: (1) embed protein sequences or multiple sequence alignments into a structured latent space using deep models (e.g., Evoformer in AF2 Jumper et al. (2021), transformer encoders in ESMFold Lin et al. (2023)); (2) predict structural outputs or downstream properties such as binding affinities or contact maps Dhakal et al. (2022); Zhang et al. (2024); and (3) evaluate or interpret the results through a combination of performance metrics and error analysis. However, benchmarks in this space often evaluate these stages in isolation, focusing either on structural accuracy (e.g., CASP Moult et al. (2018)), binary disorder classification (e.g., CAID Necci et al. (2021)), or post hoc visualizations without a unified pipeline to assess model reliability across data, task, and user-facing dimensions.

To address this gap, DisProtBench adopts a three-level benchmarking paradigm: **1. Data Level:** formalizes biologically grounded input complexity through curated datasets and unified representations. **2. Task Level:** defines model functionalities and task-specific evaluation metrics. **3. User**

**Level:** emphasizes interpretability, comparative diagnostics, and accessibility via an interactive web interface.

These levels reflect and encompass the core stages of protein modeling workflows, from data preprocessing to predictive modeling and decision support, providing a comprehensive lens for evaluating model utility under real-world biological constraints.

Figure 2: DisProtBench integrates biologically grounded datasets (Data Level), multi-perspective evaluation (Task Level), and an interactive web portal (User Level) to support rigorous, interpretable, and extensible benchmarking.

Figure 2 illustrates the architecture of DisProtBench. At the **Data Level**, diverse sources, including DisProt-annotated disorder regions, GPCR–ligand interactions, and multimeric protein complexes, are unified through automatic sanitization and standardized 3D graph representations, enabling consistent input for downstream evaluation. The **Task Level** categorizes PSPMs by input modality, structural backbone, and source, and evaluates them via an extensible toolbox. This toolbox separates structure generation (e.g., diversity, validity, efficiency) from prediction tasks (e.g., accuracy, applicability), reflecting functional deployment in PPI and drug discovery scenarios. Finally, the **User Level** provides a visual-interactive interface for model selection, error exploration, and human-AI collaboration. It supports flexible configuration, including FEM simulations and runnable demos, allowing users to trace how structural discrepancies affect downstream decisions. By unifying these levels, DisProtBench establishes a principled framework for evaluating PSPMs in biologically relevant, interpretability-aware, and application-centered contexts.

## 4 DISPROTBENCH DATABASE DEVELOPMENT

To enable robust, biologically meaningful evaluation of PSPMs, particularly for PPI and drug discovery applications, DisProtBench consolidates datasets that span key real-world dimensions highlighted in the Introduction: *(i)* structural uncertainty due to intrinsic disorder, *(ii)* interaction complexity involving multimeric assemblies or ligand specificity, and *(iii)* clinical or functional relevance (e.g., pathogenic variants, pharmacological targets). Most existing benchmarks address only a subset of these axes: CAID Necci et al. (2021) focuses on static disorder without functional context, while CASP Moult et al. (2018) emphasizes fold accuracy but overlooks disorder-mediated binding. To bridge this gap, DisProtBench integrates three curated datasets, each capturing distinct yet complementary aspects of protein modeling, enabling a comprehensive assessment of both structural fidelity and functional reliability.

**1. DisProt-Based Dataset: Disorder in Human Disease.** To evaluate PSPMs performance under structural uncertainty and functional constraint, we curate proteins with long disordered segments (≥20 residues) from the UniProt Database. We integrate this with high-confidence interaction partners from HINT and pathogenic variant annotations from ClinVar, retaining protein pairs where both genes harbor ≥2 disease-associated missense mutations. This setup evaluates whether models can preserve biologically meaningful disordered regions and correctly represent structures implicated in disease, critical for interpreting structure–function breakdowns in clinical genomics.

**2. Individual Protein Dataset: Disorder and Ligand Binding.** Drug discovery pipelines often depend on structural fidelity in binding pockets while tolerating ambiguity in flexible or unstructured domains. To assess model behavior under such conditions, we compile 71,757 GPCR–ligand pairs (top 20 GPCRs) and 33,212 pain-related pairs from ChEMBL and BindingDB, filtered for $K_i$ values. We retain potent compounds ($pK_i$ 6–9), remove duplicates using InChIKey and UniProt ID, and annotate 10,816 agonist/antagonist interactions, 2,308 FDA-approved drugs, and 379 gut metabolites Yang et al. (2024). This dataset tests PSPMs across interaction specificity (agonist vs. antagonist), structural sensitivity (via binding affinity), and relevance to pharmacologically diverse compounds, capturing the functional demands of structure-based drug design.

**3. Protein Interaction Dataset: Disorder-Mediated Interfaces.** Multimeric complexes often exhibit binding driven by flexible or transient interfaces, conditions under which traditional fold-accuracy metrics are unreliable. To test PSPMs in this setting, we construct a high-resolution PPI dataset using AlphaFold-Multimer predictions, filtered for structural quality (median pLDDT ≥70, pDockQ, interface compactness) and functional disorder (interface overlap with IDRs). This dataset challenges PSPMs to recover biologically plausible complex geometries despite the presence of disordered segments, mirroring real-world conditions in systems biology and protein engineering Hu & Ohue (2024b).

Taken together, these datasets ensure that DisProtBench comprehensively evaluates PSPMs across all three dimensions: structural robustness, functional specificity, and translational relevance. By moving beyond static fold accuracy toward biologically grounded validation, DisProtBench sets a new standard for disorder-aware, application-driven protein structure benchmarking.

## 5 DISPROTBENCH TOOLBOX DEVELOPMENT

### 5.1 PLDDT AS A PROXY FOR INTRINSIC DISORDER

Benchmarking PSPMs under biological uncertainty is challenging due to limited experimentally validated IDR annotations. Recent studies show that AlphaFold's per-residue confidence score (pLDDT) often correlates with structural ambiguity and flexibility Tunyasuvunakool et al. (2021); Wilson et al. (2022), with low pLDDT scores frequently marking disordered regions. However, this association has largely remained qualitative. Building on these observations, we investigate whether pLDDT can be used as a quantitative and scalable proxy for IDRs in benchmarking settings. Specifically, we assess whether low-confidence residues (pLDDT < 50) align with known disorder segments and enable consistent evaluation in the absence of ground-truth labels.

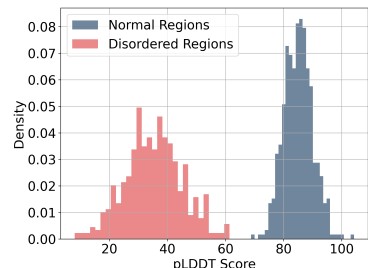

Figure 3: Distribution of pLDDT scores for ordered and disordered residues.

We conduct an empirical analysis using our DisProt-based dataset, which includes residue-level disorder annotations supported by experimental evidence. For each residue, we extract the pLDDT score from AF3 predictions and stratify the distribution by disorder label. As shown in Figure 3, pLDDT scores follow a clear bimodal distribution: disordered residues cluster around 30–40, while ordered residues peak above 85, with minimal overlap. This strong separation validates the use of pLDDT as a proxy for structural uncertainty and a disorder-aware surrogate. Based on this insight, DisProtBench adopts pLDDT-based stratification to evaluate model performance under different confidence regimes. Specifically, we report downstream task results using three thresholds—full sequence, pLDDT ≥ 30, and pLDDT ≥ 50—allowing us to assess model robustness in ambiguous

regions and extend evaluation to datasets lacking explicit IDR annotations. Our choice of pLDDT is consistent with the previously established paradigm (Aspromonte et al., 2024; Abramson et al., 2024b; Piovesan et al., 2022); details have been provided in Appendix 7.

## 5.2 EXPERIMENTAL SETTING

Building on Section 5.1, which establishes pLDDT as a proxy for structural uncertainty and intrinsic disorder, we stratify evaluation into three confidence levels: full sequence, pLDDT $\geq$30, and pLDDT $\geq$50. This stratification enables us to assess PSPM robustness in both well-structured and ambiguous regions.

**Tasks.** We evaluate models on two representative biomedical applications: **(1) Protein–Protein Interaction (PPI) prediction**, which often involves flexible, disorder-mediated interfaces; **(2) Drug Discovery**, an application task of protein ligand-binding disorder affinity prediction.

**Baselines.** We benchmark eleven PSPMs spanning diverse design choices: AlphaFold2, AlphaFold-Multimer, ESM-2, ESMFold, ProtBERT, ProtT5, RoseTTAFold, OmegaFold, HelixFold, FastFold, and OpenFold. These models differ in architecture (Evoformer, Transformer, Hybrid), input modality (MSA-based vs. sequence-only), and resolution (atomic vs. coarse-grained).

**Evaluation Metrics.** We adopt a unified evaluation framework comprising: **(1) Classification metrics:** Accuracy, Precision, Recall, and F1; **(2) Regression metrics:** Mean Absolute Error (MAE), Mean Squared Error (MSE), and Pearson correlation ($R$); **(3) Structure-aware interface metrics:** Receptor Precision/Recall (PP, PR) and Ligand Precision/Recall (LP, LR)(as shwon in Table 4). Formal definitions of all metrics are provided in the Appendix 7.

This design enables a comprehensive comparison of PSPMs across tasks, evaluation criteria, and structural uncertainty regimes.

## 5.3 EXPERIMENTAL RESULTS AND ANALYSIS

Building on our findings in Section 5.1, we evaluate how PSPMs behave across two downstream tasks, PPI prediction and ligand-binding affinity estimation, under varying structural confidence. These experiments aim to test whether PSPMs can maintain functional reliability when exposed to disorder-prone regions. By stratifying evaluations using pLDDT thresholds, we analyze how intrinsic disorder affects both structure quality and application-level utility.

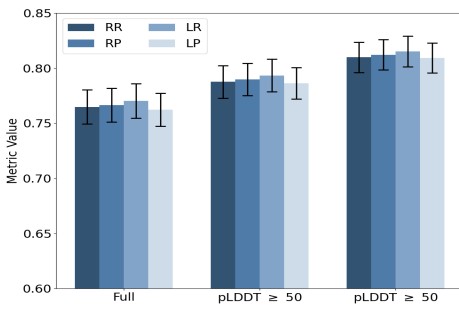
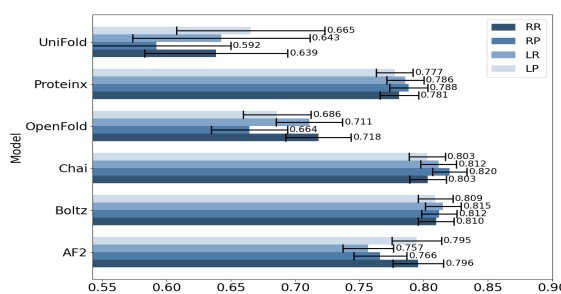

(a) RR, LR, RP, and LP of Boltz under varying pLDDT thresholds with its 95% CI.

(b) RR, LR, RP and LP across PSPMs under pLDDT $\geq$ 50 with its 95% CI.

Figure 4: Performance comparison of Receptor and Ligand Precision/Recall metrics on predicted structure with 95% CI.

**PPI Prediction Performance.** We examine the robustness of PSPMs in predicting PPI, a setting where disordered regions frequently mediate transient or flexible binding. Figure 4 summarizes interface recovery metrics (RR, RP, LR, LP) with 95% confidence intervals. As shown in Figure 4a, filtering out residues with low pLDDT ($<$ 50) substantially improves Boltz's performance across all metrics, highlighting that IDRs introduce structural variability that impairs interpretation(Figure 4a).

Similarly, Chai and Boltz outperform other PSPMs under high-confidence constraints, followed by AF2, Proteinx, OpenFold, and UniFold in rank(Figure 4b).

Table 1: Performance comparison on PPI prediction across PSPMs under different pLDDT thresholds.

| PSPM | Original | | | | pLDDT ≥ 30 | | | | pLDDT ≥ 50 | | | |
|---|---|---|---|---|---|---|---|---|---|---|---|---|
| | Acc | Prec | Rec | F1 | Acc | Prec | Rec | F1 | Acc | Prec | Rec | F1 |
| AF2 | 0.793 | 0.783 | 0.799 | 0.791 | 0.802 | 0.791 | 0.812 | 0.801 | 0.818 | 0.809 | 0.825 | 0.817 |
| AF3 | 0.9 | 0.888 | 0.915 | 0.901 | 0.905 | 0.893 | 0.905 | 0.906 | 0.913 | 0.8989 | 0.93 | 0.914 |
| Boltz | 0.850 | 0.848 | 0.853 | 0.850 | 0.858 | 0.853 | 0.863 | 0.858 | 0.869 | 0.870 | 0.868 | 0.869 |
| Chai | 0.850 | 0.841 | 0.863 | 0.852 | 0.858 | 0.847 | 0.873 | 0.860 | 0.869 | 0.857 | 0.887 | 0.871 |
| OpenFold | 0.624 | 0.605 | 0.605 | 0.605 | 0.643 | 0.622 | 0.638 | 0.630 | 0.671 | 0.656 | 0.651 | 0.653 |
| Proteinx | 0.810 | 0.809 | 0.812 | 0.810 | 0.819 | 0.820 | 0.818 | 0.819 | 0.834 | 0.834 | 0.835 | 0.834 |
| UniFold | 0.552 | 0.378 | 0.667 | 0.483 | 0.567 | 0.389 | 0.667 | 0.491 | 0.597 | 0.417 | 0.714 | 0.526 |

To quantify these differences in task-level accuracy, we evaluate precision, recall, and F1 scores for each model at three confidence levels(Table 1). Removing disordered residues consistently improves F1 scores, up to +0.04 in some cases, confirming the disruptive effect of IDRs. AF3 leads overall with F1 = 0.914, while Boltz, Chai, and Proteinx cluster closely behind. OpenFold and UniFold show persistent deficits even in high-confidence regions. These outcomes are statistically validated by pairwise McNemar tests (Figure 5), where AF3 significantly outperforms most PSPMs $(-\log_{10} p \geq 10)$, and the Boltz-Chai-Proteinx group demonstrates resilience to structural uncertainty. These findings confirm our methodological claim: IDRs must be explicitly accounted for to assess model generalizability and PPI prediction accuracy.

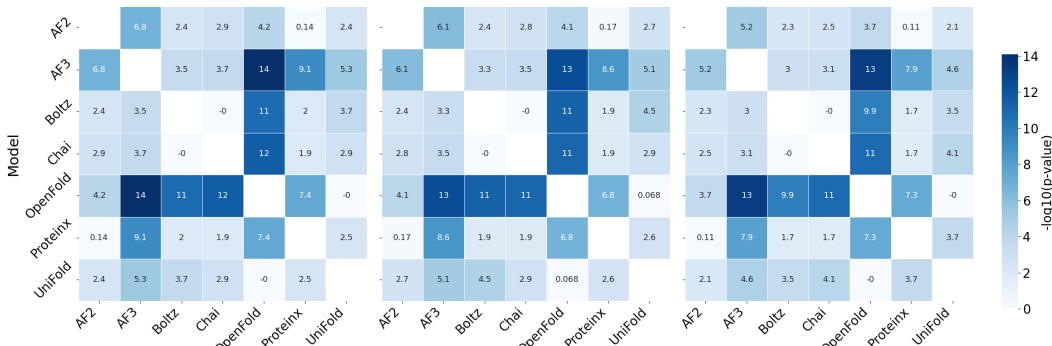

Figure 5: Heatmaps of $-\log_{10}(p)$ values from McNemar tests comparing pairwise model performance on PPI prediction across different pLDDT thresholds. Left: full sequence; Middle: pLDDT $\geq$ 30; Right: pLDDT $\geq$ 50. Higher values indicate more statistical significance between PSPMs. Blank blocks indicate self-comparisons, which are omitted by definition.

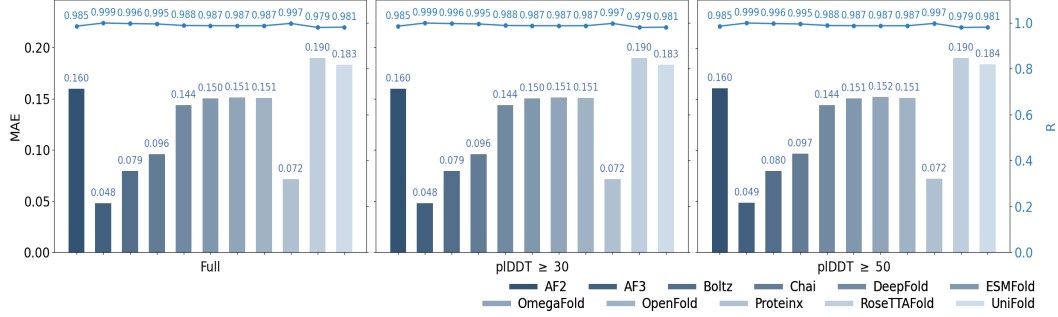

Figure 6: Performance comparison of MAE and $R$ on drug discovery prediction across PSPMs under different pLDDT thresholds.

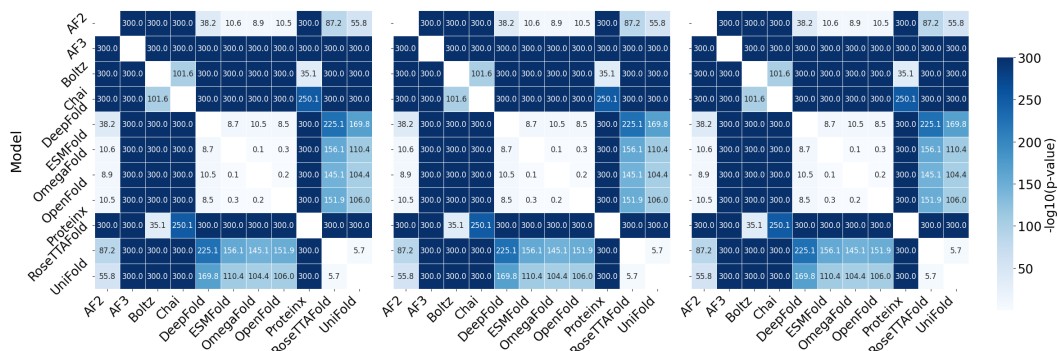

Figure 7: Heatmaps of $-\log_{10}(p)$ values from Wilcoxon signed-rank tests comparing model performance in drug discovery tasks across different pLDDT thresholds. Left: full sequence; Middle: pLDDT $\geq$ 30; Right: pLDDT $\geq$ 50. Higher values indicate greater statistical significance in pairwise differences between PSPMs. Blank blocks indicate self-comparisons, which are omitted by definition.

**Drug Discovery Performance.** In contrast, we observe that structure-based drug discovery tasks are relatively robust to IDRs. Figure 6 presents MAE and $R$ scores across PSPMs and pLDDT thresholds. AF3 remains the highest rank (MAE = 0.048, $R \approx 0.999$), with Boltz, Chai, and Proteinx maintaining performance across thresholds. Notably, model rankings remain stable regardless of disorder filtering, suggesting that drug-binding affinity prediction primarily depends on well-structured binding pockets rather than disordered interfaces. This resilience is further confirmed by Wilcoxon signed-rank tests (Figure 7), which show significant gaps between top-tier and weaker PSPMs (i.e., UniFold, OpenFold, RoseTTAFold), but little sensitivity to confidence thresholding.

**Key Insights.** Together, our benchmark suggests three conclusions. First, confidence-based filtering (via pLDDT) serves as a useful mechanism for isolating the functional impact of disordered regions on the predicted protein structures. Second, disorder disrupts tasks involving extended interaction interfaces (e.g., PPIs) far more than localized ones (e.g., ligand binding). Third, evaluation fidelity hinges on stratifying PSPMs by structural certainty, without which global metrics obscure model brittleness in complex biological contexts. These insights motivate the need for disorder-aware evaluation protocols and justify the three-tiered structure of DisProtBench. Full results and additional ablations are provided in the Appendix.

## 6   DISPROTBENCH VISUAL-INTERACTIVE INTERFACE

To support real-world deployment of protein structure prediction models, aggregate accuracy metrics alone are insufficient. Researchers need tools for localized error analysis, model comparison, and structure–function interpretation. As a key contribution, we introduce a visual-interactive web portal that enhances interpretability and benchmark usability. Unlike prior portals (e.g., CASP Moult et al. (2018), CAID Necci et al. (2021); Conte et al. (2023)), which provide limited access to structure or disorder data, our portal integrates structural uncertainty, downstream task performance, and model-specific variation in a unified interface. Users can explore confidence-stratified predictions, visualize model outputs, and assess functional consequences of structural differences. Full instructions are available on our website.

The visual-interactive portal (Figure 8) is organized into six key components that reflect typical workflows in structure-function evaluation. Users begin by selecting a task type, PPI prediction, drug discovery, or server testing, via the navigation panel (a), then specify input via protein ID or sequence (b), supporting both database queries and custom evaluations. A PSPM selection panel (c) enables users to choose multiple prediction PSPMs for side-by-side analysis, facilitating model comparison under varying structural assumptions and architectural designs. Structural similarity is visualized through an RMSD-based alignment heatmap (d), helping identify discrepancies across model outputs, particularly in disordered regions. To enable detailed exploration, the visualization panel (e) renders 3D protein structures with B-factor coloring and alignment overlays, revealing uncertainty and variation at residue-level resolution. Finally, the downstream prediction panel (f) links structural

inputs to biological tasks by displaying functional predictions from each model, highlighting how IDRs and confidence scores influence real-world applicability. Together, these features lead to a powerful tool for error analysis, model diagnosis, and deployment readiness evaluation.

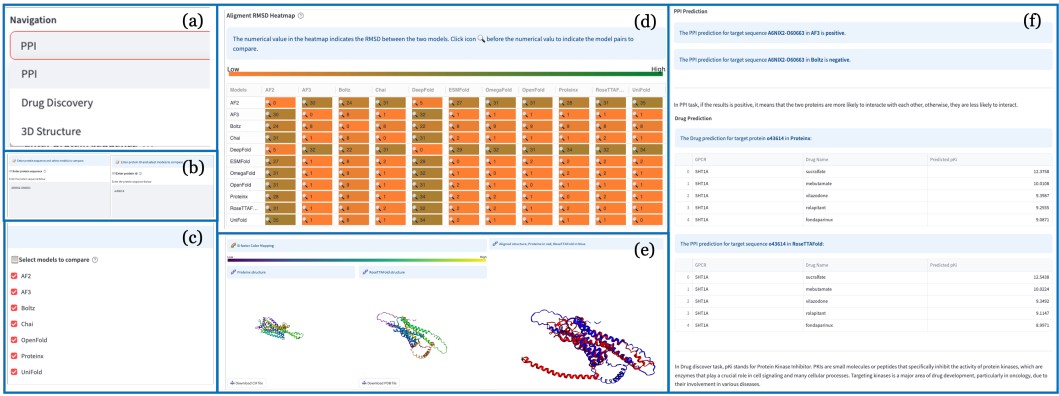

Figure 8: Portal Overview. (a) Navigation for task type. (b) Protein input. (c) Model selection. (d) Structural alignment heatmap. (e) 3D visualization and residue-level uncertainty. (f) Downstream task predictions.

Overall, the DisProtBench portal provides an integrated environment for structure-function benchmarking, unifying structural comparison, disorder visualization, and task-level analysis in a single tool. By making disorder-aware evaluation transparent and interactive, it helps the community bridge the gap between predictive accuracy and biological utility, accelerating model development.

## 7 CONCLUSION

We present DisProtBench, a comprehensive benchmark for evaluating the functional reliability of PSPMs, with a focus on biologically realistic, disorder-enriched scenarios. Our contributions include: (1) curated datasets spanning PPI, ligand binding, and intrinsic disorder; (2) a modular evaluation toolbox for analyzing structure–function relationships across multiple metrics and confidence levels; and (3) an interactive web portal enabling model inspection and comparison. Through systematic experiments, we demonstrate that intrinsic disorder—often overlooked—significantly impacts structure prediction and downstream tasks. While models like AF3 show robust performance, others struggle under structural uncertainty, highlighting the need for IDR-aware evaluation. Notably, we find that low-confidence regions (e.g., pLDDT $< 50$) correlate with functional errors in PPI prediction, whereas drug discovery tasks, which rely on structured pockets, are more robust—underscoring the value of task-specific modeling.

DisProtBench helps bridge the gap between predictive accuracy and biological utility, encouraging evaluation protocols that are both statistically rigorous and biologically meaningful. We recommend future efforts to: (1) develop architectures that explicitly handle low-confidence or disordered regions; (2) integrate functional prediction into end-to-end model training; (3) adopt uncertainty-aware training strategies; and (4) create specialized benchmarks for underexplored tasks like phase separation and disorder-mediated interactions. Models should also report stratified performance by confidence levels, as aggregate metrics may mask vulnerabilities in critical regions. Future developments of DisProtBench include support for coarse-grained/ensemble predictions Shrestha et al. (2021), alternative disorder annotations Piovesan et al. (2023), and real-time interpretability tools Medina-Ortiz et al. (2024). While DisProtBench enhances transparency and utility, careful deployment remains essential, especially in clinical contexts where mischaracterizing disordered regions can lead to harmful outcomes. We envision DisProtBench as a foundation for advancing robust, biologically grounded structure prediction.

## REPRODUCIBILITY STATEMENT

We ensure the reproducibility of our benchmark through multiple measures. All evaluated PSPMs are publicly available online and referenced in the Appendix. Detailed descriptions of task setups, data preprocessing, and evaluation protocols are provided in Section 7, with metric definitions included in the Appendix. Hyperparameters and stratification criteria (pLDDT, actifpTM, PAE) are explicitly documented, and all results are reported under consistent settings for classification, regression, and structure-aware metrics. To further facilitate replication, we release scripts for data processing and evaluation, along with the benchmark datasets, at: DisProtBench. Collectively, these resources enable independent verification and extension of our results.

## ETHICAL STATEMENT

Our benchmark dataset is constructed from publicly available, well-curated biological resources such as PDB and UniProt, which comply with established ethical standards for data sharing. No web-scraped or user-generated content is included, thereby reducing risks associated with sensitive or unvetted data. To promote transparency while encouraging responsible use, we release the dataset under the MIT License, which permits open academic use while placing the responsibility of ethical application on the user. We also provide clear documentation outlining the intended research use cases and explicitly discourage any misuse, including the generation or manipulation of biological sequences for non-research purposes. No generative models trained on this dataset are distributed.

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

APPENDIX

## 1. EVALUATION SETUP

To ensure fair and consistent comparison across protein structure prediction models (PSPMs), we designed a unified evaluation framework that spans three key dimensions: data standardization, metric consistency, and confidence-aware stratification. All models were assessed on the same biologically grounded datasets curated to reflect real-world complexity—encompassing disordered regions, ligand-bound G-protein-coupled receptors (GPCRs), and multimeric complexes with interface ambiguity. These datasets were preprocessed using a common pipeline to ensure uniformity in input representations, such as standardized 3D graph formats.

We employed a consistent set of evaluation metrics across tasks, including classification metrics (Accuracy, Precision, Recall, F1), regression metrics (Mean Absolute Error (MAE), Mean Squared Error (MSE), Pearson correlation $R$), and structure-aware interface scores (Receptor/Ligand Precision and Recall). To account for structural uncertainty, we stratified all evaluations by per-residue pLDDT scores using three thresholds—full sequence, predicted Local Distance Difference Test (pLDDT) $\geq$ 30, and pLDDT $\geq$ 50—enabling systematic analysis of model robustness under varying confidence levels.

In addition, we developed an extensible benchmarking toolbox to automate evaluation across these settings and ensure reproducibility. This setup is further complemented by the DisProtBench Portal, which provides interactive, side-by-side visualizations of model outputs and error patterns under identical conditions. Together, these components form a unified and transparent framework that supports biologically meaningful comparisons across diverse PSPMs and tasks.

## 2. OVERVIEW OF BENCHMARKED MODEL ARCHITECTURES

We benchmark eleven PSPMs and list the details of each model used in our benchmark. Each model is evaluated under uniform settings for protein–protein interaction (PPI) and/or drug discovery prediction tasks. This benchmarking effort specifically investigates how different model architectures perform in the presence of Intrinsically Disordered Regions (IDRs), which pose unique challenges for structural prediction and are critical for understanding flexible, functionally important protein interfaces.

- **AlphaFold2 (AF2)** Jumper et al. (2021), developed by DeepMind, is built on the Evoformer architecture and structure module. The Evoformer processes both MSA and pairwise residue representations using a combination of axial attention, outer product mean, and triangle multiplication updates, enabling the modeling of long-range evolutionary dependencies. The structure module then predicts 3D atomic coordinates via invariant point attention and torsion angle regression. This architecture has set the standard for atomic-level protein structure prediction.

- **OpenFold** Ahdritz et al. (2024), developed by AQLab, is a faithful reimplementation of AF2 in PyTorch, retaining the Evoformer backbone and model design. While architecturally aligned with AF2, OpenFold emphasizes modularity and accessibility for the research community, enabling flexible adaptation and experimentation.

- **UniFold** Li et al. (2022), introduced by DeepModeling, also replicates the AF2 architecture but incorporates computational optimizations for scalability. It supports distributed training and batched inference, allowing efficient deployment on large-scale protein datasets while maintaining the original Evoformer-based modeling capabilities.

- **AlphaFold3 (AF3)** Abramson et al. (2024a), the next-generation model from DeepMind, extends the Evoformer with a pretrained protein language model (LLM), enabling the integration of both MSA-derived and single-sequence embeddings. This multimodal approach enhances the model's ability to generalize to MSA-scarce proteins. Additionally, AF3 introduces modules for ligand-aware modeling, supporting the prediction of protein-ligand complexes with atomic resolution. The model accepts both sequence and MSA inputs and outputs include 3D structures of proteins and bound ligands.

- **Boltz** Wohlwend et al. (2024), developed by the MIT Jameel Lab, is a transformer-based model that operates solely on sequence inputs. Its architecture consists of standard multi-

head self-attention layers, trained with masked language modeling and supervised by structural signals such as contact maps. It is particularly suited for modeling flexible and disordered protein regions and produces coarse-grained structural outputs.

- **Chai**Chai Discovery (2024), from Chai Discovery, uses a lightweight transformer architecture optimized for drug discovery. It focuses on capturing key functional sites with efficient attention mechanisms and positional encodings tailored to domain boundaries. The model operates on single-sequence inputs and is designed for rapid evaluation in therapeutic settings.

- **Proteinix**Chen et al. (2025), introduced by ByteDance, enhances the transformer backbone with ligand-aware capabilities. It integrates structural cues from ligands during training and performs joint modeling of protein structures and ligand binding. The model outputs both atomic protein structures and ligand poses, enabling end-to-end prediction of molecular complexes.

- **ESMFold**Lin et al. (2023), released by Meta AI, combines large-scale protein language modeling with a folding head to generate 3D structures. It uses the ESM-2 model to produce contextualized residue embeddings from single sequences, bypassing the need for MSA. These embeddings are then passed to a structure module that outputs backbone and side-chain coordinates. ESMFold offers exceptional speed and is especially useful for predicting structures of proteins without homologs.

- **OmegaFold**Wu et al. (2022), developed by HeliXon, uses a transformer-based encoder to embed sequence inputs and predict 3D coordinates through geometric vector outputs and torsion angle heads. It is designed for fast, MSA-free inference and supports coarse-grained output suitable for high-throughput applications in structural biology and drug discovery.

- **RoseTTAFold**Baek et al. (2021), developed by the Baker Lab, employs a hybrid 3-track network that simultaneously processes 1D sequence features, 2D residue-residue relationships, and 3D coordinates. It combines convolutional layers for capturing local sequence context with attention mechanisms that model long-range interactions. Cross-track communication enables holistic structure learning, and the model predicts full atomic structures using MSA inputs. RoseTTAFold offers a fast alternative to AF2 with broad utility across structural tasks.

- **DeepFold**Lee et al. (2023), from Hanyang University, adopts a custom deep learning architecture comprising 1D convolutional layers, fully connected modules, and a geometric decoder. It operates solely on single-sequence inputs and is optimized for atomic-level predictions in drug-related tasks. The design emphasizes speed and simplicity while maintaining sufficient accuracy for practical applications in structure-based screening.

3. Downstream Applications and Evaluation Tasks

We evaluate each PSPM on two key downstream tasks: PPI prediction and drug binding affinity estimation. These tasks require accurate structural modeling, particularly in intrinsically disordered regions, and allow us to assess real-world utility beyond static structure accuracy.

**1. PPI Prediction** The pipeline begins with a predicted protein complex structure generated by AlphaFold-Multimer from a pair of input sequences. The resulting PDB file is used as input to extract the protein–protein interface, which is then encoded using one-hot, volumetric, or distance-based representations. This encoded tensor is passed through a neural network, where data augmentation is applied. Finally, a backbone architecture—either DenseNet or ResNet—predicts the probability that the input proteins interact(as shown in Figure 9).

**2. Drug Discovery** The pipeline begins with a predicted protein–ligand complex structure, which serves as input to two parallel modules. First, the molecular encoder processes the ligand component to extract latent chemical features. Simultaneously, the protein component is passed through an Evoformer module to derive a structured protein representation. These two feature sets are then concatenated to form a unified representation of the protein–ligand interaction. This combined feature vector is subsequently fed into the LISA-CPI model, which outputs a ranked list of predicted activity scores, with the top 5 scores indicating the most likely compound–protein interactions(as shown in Figure 10).

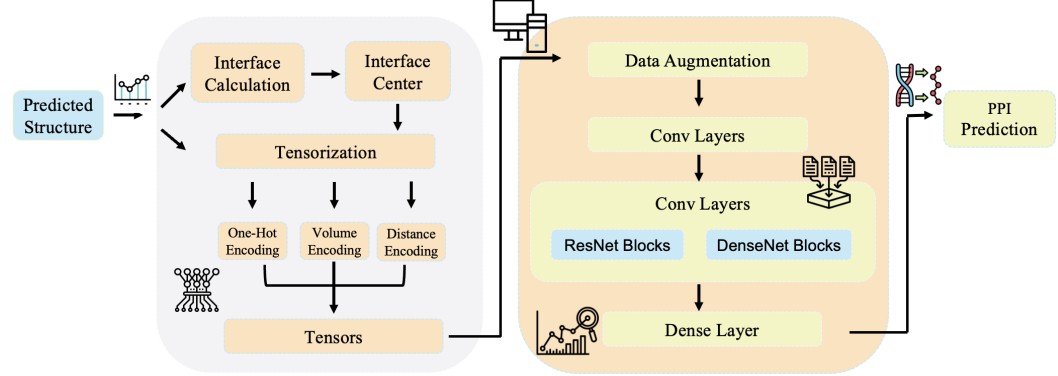

Figure 9: Overview of PPI Prediction

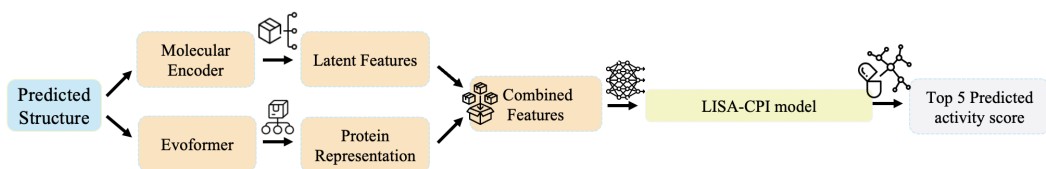

Figure 10: Overview of Drug Discovery

## 4. DISPROTBENCH EXPERIMENTAL DESIGN AND EVALUATION FRAMEWORK

To ensure robust and comprehensive evaluation, we conduct an extensive number of experiments within DisProtBench, our curated benchmark designed to reflect the structural complexity introduced by IDRs. By systematically varying input conditions, evaluation metrics, and task settings across diverse proteins, we capture fine-grained differences in model performance that may be obscured in simpler benchmarks. This breadth of experimentation enables us to rigorously assess model generalizability, sensitivity to disorder, and consistency across downstream applications, providing a more realistic and informative portrait of PSPM capabilities in challenging biological scenarios.

### 4.1 DEFINITIONS AND ROLES OF LIGAND AND RECEPTOR

We follow the definitions of ligand and receptor in Verburgt et al. (2022). The following definitions are used to compute structural interface scores, enabling consistent and biologically grounded evaluation across our benchmark.

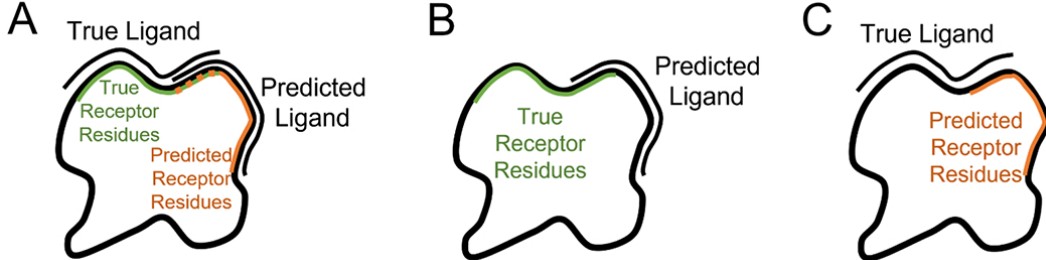

Figure 11: Diagrams depicting the interface residue metrics. A, Visualization of interfaces used in the Receptor Precision and Receptor Recall metrics. B, Visualization of the ligand and receptor residues used in the Ligand Precision metric. C, Visualization of the ligand and receptor residues used in the Ligand Recall metric.

## 4.2 EVALUATION METRICS (CLASSIFICATION, REGRESSION, AND STRUCTURAL)

To comprehensively assess model performance across diverse interaction scenarios, we employ a suite of evaluation metrics that capture both predictive accuracy and structural fidelity. These metrics span classification, regression, and structure-aware categories, enabling robust comparison across tasks such as protein–protein interaction prediction and ligand binding affinity estimation. Below, we define each metric used in our benchmark and outline its relevance to the specific evaluation context.

Table 2: Classification Metrics

| Metric | Definition / Formula |
|---|---|
| Precision (Positive Predictive Value) | $\dfrac{\text{TP}}{\text{TP} + \text{FP}}$ |
| Recall (Sensitivity) | $\dfrac{\text{TP}}{\text{TP} + \text{FN}}$ |
| F1 Score | $\dfrac{2 \cdot \text{TP}}{2 \cdot \text{TP} + \text{FP} + \text{FN}}$ |
| Accuracy | $\dfrac{\text{TP} + \text{TN}}{\text{TP} + \text{TN} + \text{FP} + \text{FN}}$ |

Table 3: Regression Metrics

| Metric | Definition / Formula |
|---|---|
| MAE | $\dfrac{1}{N} \sum_{i=1}^{N} |y_i - \hat{y}_i|$ |
| MSE | $\dfrac{1}{N} \sum_{i=1}^{N} (y_i - \hat{y}_i)^2$ |
| Pearson $R$ | $\dfrac{\sum_{i=1}^{N}(y_i - \bar{y})(\hat{y}_i - \bar{\hat{y}})}{\sqrt{\sum_{i=1}^{N}(y_i - \bar{y})^2} \cdot \sqrt{\sum_{i=1}^{N}(\hat{y}_i - \bar{\hat{y}})^2}}$ |

Table 4: Structural Interface Metrics

| Metric | Definition / Formula |
|---|---|
| Receptor Precision (RP) | $\dfrac{|\text{True Receptor Interface} \cap \text{Predicted Receptor Interface}|}{|\text{Predicted Receptor Interface}|}$ |
| Receptor Recall (RR) | $\dfrac{|\text{True Receptor Interface} \cap \text{Predicted Receptor Interface}|}{|\text{True Receptor Interface}|}$ |
| Ligand Precision (LP) | $\dfrac{|\text{Predicted Ligand Interface} \cap \text{True Receptor Interface}|}{|\text{Predicted Ligand Interface}|}$ |
| Ligand Recall (LR) | $\dfrac{|\text{True Ligand Interface} \cap \text{Predicted Receptor Interface}|}{|\text{True Ligand Interface}|}$ |

## 4.3 PERFORMANCE ANALYSIS ACROSS STRUCTURAL CONFIDENCE REGIMES

To assess how PSPMs perform under different levels of structural confidence, we stratify our evaluation across three regimes: full sequence (no pLDDT filtering), moderately structured regions (pLDDT $\geq$ 30), and highly structured regions (pLDDT $\geq$ 50). This setup reflects increasing exclusion of IDRs, allowing us to systematically analyze how model predictions shift as structural ambiguity decreases.

**Comparison of Structural Prediction** Across all thresholds, we observe improved model performance with increasing structural confidence, indicating that IDRs pose significant challenges

for accurate interface prediction. Boltz and Chai consistently achieve strong RR/RP/LR/LP (receptor/ligand recall, receptor/ligand precision, respectively) scores across all regimes, demonstrating robustness even in disordered contexts. In contrast, OpenFold and UniFold show substantial performance drops on full-sequence evaluations, with notable recovery only in highly structured subsets, suggesting greater reliance on well-defined backbone geometry. These findings emphasize the need for PSPMs to explicitly handle disorder-aware scenarios, as IDRs frequently mediate biologically critical interactions that static confidence metrics alone may not capture.

Table 5: RR/RP/LR/LP on full sequences (Original), reported as mean $\pm$ 95% CI.

| Model | RR | RP | LR | LP |
|---|---|---|---|---|
| AF2 | $0.749 \pm 0.0217$ | $0.7186 \pm 0.0226$ | $0.7052 \pm 0.0215$ | $0.7486 \pm 0.0215$ |
| Boltz | $0.7648 \pm 0.0155$ | $0.7666 \pm 0.0153$ | $0.7703 \pm 0.0155$ | $0.7624 \pm 0.0140$ |
| Chai | $0.7583 \pm 0.0158$ | $0.7746 \pm 0.0149$ | $0.7674 \pm 0.0155$ | $0.7571 \pm 0.0155$ |
| OpenFold | $0.6626 \pm 0.0271$ | $0.6127 \pm 0.0312$ | $0.6551 \pm 0.0274$ | $0.6322 \pm 0.0280$ |
| Proteinx | $0.7344 \pm 0.0164$ | $0.7427 \pm 0.0162$ | $0.7385 \pm 0.0159$ | $0.731 \pm 0.0159$ |
| UniFold | $0.5876 \pm 0.0593$ | $0.5431 \pm 0.0604$ | $0.5892 \pm 0.0711$ | $0.6116 \pm 0.0609$ |

Table 6: RR/RP/LR/LP on structured regions (pLDDT $\geq$ 30), reported as mean $\pm$ 95% CI.

| Model | RR | RP | LR | LP |
|---|---|---|---|---|
| AF2 | $0.7729 \pm 0.0208$ | $0.7426 \pm 0.0217$ | $0.7313 \pm 0.0207$ | $0.7719 \pm 0.0204$ |
| Boltz | $0.7876 \pm 0.0147$ | $0.7899 \pm 0.0146$ | $0.7934 \pm 0.0148$ | $0.7863 \pm 0.0142$ |
| Chai | $0.7815 \pm 0.015$ | $0.798 \pm 0.0142$ | $0.7898 \pm 0.0147$ | $0.7804 \pm 0.0148$ |
| OpenFold | $0.6904 \pm 0.0263$ | $0.6386 \pm 0.0306$ | $0.6828 \pm 0.0267$ | $0.6593 \pm 0.0272$ |
| Proteinx | $0.7584 \pm 0.0158$ | $0.7658 \pm 0.0155$ | $0.7626 \pm 0.0152$ | $0.7544 \pm 0.0155$ |
| UniFold | $0.6127 \pm 0.0576$ | $0.5671 \pm 0.059$ | $0.6172 \pm 0.0707$ | $0.6384 \pm 0.0595$ |

Table 7: RR/RP/LR/LP on highly structured regions (pLDDT $\geq$ 50), reported as mean $\pm$ 95% CI.

| Model | RR | RP | LR | LP |
|---|---|---|---|---|
| AF2 | $0.7959 \pm 0.0197$ | $0.7663 \pm 0.0207$ | $0.7569 \pm 0.0198$ | $0.7946 \pm 0.0193$ |
| Boltz | $0.8098 \pm 0.0139$ | $0.8121 \pm 0.0138$ | $0.8152 \pm 0.014$ | $0.8093 \pm 0.0134$ |
| Chai | $0.8034 \pm 0.0142$ | $0.8203 \pm 0.0134$ | $0.8117 \pm 0.0139$ | $0.8029 \pm 0.014$ |
| OpenFold | $0.7182 \pm 0.0254$ | $0.6644 \pm 0.0297$ | $0.7111 \pm 0.0258$ | $0.6861 \pm 0.0263$ |
| Proteinx | $0.7811 \pm 0.015$ | $0.7884 \pm 0.0147$ | $0.7857 \pm 0.0144$ | $0.7775 \pm 0.0144$ |
| UniFold | $0.6386 \pm 0.0557$ | $0.5923 \pm 0.0576$ | $0.6426 \pm 0.0689$ | $0.6655 \pm 0.0578$ |

**MSE of Training Data on Drug Discovery**    Table 8 presents the mean squared error (MSE) of drug discovery predictions across PSPMs under varying pLDDT thresholds. Most models show minimal MSE variation as the threshold increases, indicating that structured regions (pLDDT $\geq$ 30 or $\geq$ 50) do not substantially alter performance. However, this stability may mask deficiencies in handling IDRs, which are excluded at higher thresholds. Notably, models like AF3, Boltz, and Chai achieve low MSE even on full sequences, suggesting effective handling of disordered regions. In contrast, RoseTTAFold and UniFold exhibit relatively high MSEs that persist regardless of pLDDT filtering, indicating limited adaptability to both structured and disordered contexts. These results highlight the importance of evaluating PSPMs across disorder-sensitive regimes to uncover their ability to generalize in real-world drug discovery scenarios, where IDRs often play regulatory and binding roles.

## 4.4 SUITABILITY OF pLDDT STRATIFICATION

Our use of pLDDT stratification is motivated both by our own analysis in Figure 3 and by the prior established paradigm (Aspromonte et al., 2024; Abramson et al., 2024b; Piovesan et al., 2022). Our extended evaluations of alternative indicators—actifpTM (Varga et al., 2025), PAE, and their combinations with pLDDT(Table 9-12)-show that all **underperform** using pLDDT alone, suggesting that pLDDT already captures the dominant signal associated with disordered regions.

Table 8: Performance comparison of MSE on drug discovery prediction of training data across PSPMs under different pLDDT thresholds.

| Model | Original) | pLDDT $\geq 30$ | pLDDT $\geq 50$ |
|---|---|---|---|
| AF2 | 0.0324 | 0.0322 | 0.0322 |
| AF3 | 0.0049 | 0.0047 | 0.004 |
| Boltz | 0.0049 | 0.0043 | 0.0042 |
| Chai | 0.0063 | 0.006 | 0.0063 |
| DeepFold | 0.0288 | 0.0282 | 0.0281 |
| ESMFold | 0.0485 | 0.048 | 0.0485 |
| OmegaFold | 0.0225 | 0.0225 | 0.0219 |
| OpenFold | 0.0257 | 0.0249 | 0.0251 |
| Proteinx | 0.0168 | 0.0163 | 0.0159 |
| RoseTTAFold | 0.068 | 0.0678 | 0.0674 |
| UniFold | 0.0529 | 0.0528 | 0.0521 |

Table 9: PPI performance under different confidence thresholds: actifpTM and PAE.

| PSPM | actifpTM $\geq 0.7$ | | | | actifpTM $\geq 0.9$ | | | | PAE $< 15$Å | | | | PAE $< 10$Å | | | |
|---|---|---|---|---|---|---|---|---|---|---|---|---|---|---|---|---|
| | Acc | Prec | Rec | F1 | Acc | Prec | Rec | F1 | Acc | Prec | Rec | F1 | Acc | Prec | Rec | F1 |
| AF2 | 0.79 | 0.78 | 0.80 | 0.79 | 0.80 | 0.79 | 0.81 | 0.80 | 0.79 | 0.78 | 0.80 | 0.79 | 0.80 | 0.78 | 0.80 | 0.80 |
| AF3 | 0.89 | 0.88 | 0.89 | 0.90 | 0.90 | 0.89 | 0.90 | 0.91 | 0.89 | 0.88 | 0.89 | 0.90 | 0.90 | 0.88 | 0.90 | 0.90 |
| Boltz | 0.85 | 0.84 | 0.85 | 0.85 | 0.86 | 0.85 | 0.86 | 0.86 | 0.85 | 0.84 | 0.85 | 0.85 | 0.86 | 0.84 | 0.86 | 0.86 |
| Chai | 0.85 | 0.84 | 0.86 | 0.85 | 0.86 | 0.85 | 0.87 | 0.86 | 0.85 | 0.84 | 0.86 | 0.85 | 0.86 | 0.84 | 0.86 | 0.86 |
| OpenFold | 0.63 | 0.61 | 0.63 | 0.62 | 0.64 | 0.62 | 0.64 | 0.63 | 0.63 | 0.61 | 0.63 | 0.62 | 0.64 | 0.62 | 0.64 | 0.62 |
| Proteinx | 0.81 | 0.81 | 0.81 | 0.81 | 0.82 | 0.82 | 0.82 | 0.82 | 0.81 | 0.81 | 0.81 | 0.81 | 0.82 | 0.82 | 0.82 | 0.82 |
| UniFold | 0.56 | 0.38 | 0.66 | 0.48 | 0.57 | 0.39 | 0.67 | 0.49 | 0.56 | 0.38 | 0.66 | 0.48 | 0.56 | 0.38 | 0.66 | 0.48 |

Table 10: PPI performance under combined structural confidence filters: pLDDT$\geq 50$, actifpTM $\geq 0.9$, and PAE$<10$Å.

| PSPM | pLDDT & actifpTM | | | | actifpTM & PAE | | | | pLDDT & PAE | | | | pLDDT & PAE & actifpTM | | | |
|---|---|---|---|---|---|---|---|---|---|---|---|---|---|---|---|---|
| | Acc | Prec | Rec | F1 | Acc | Prec | Rec | F1 | Acc | Prec | Rec | F1 | Acc | Prec | Rec | F1 |
| AF2 | 0.80 | 0.79 | 0.81 | 0.80 | 0.80 | 0.78 | 0.80 | 0.80 | 0.80 | 0.78 | 0.80 | 0.80 | 0.80 | 0.78 | 0.80 | 0.80 |
| AF3 | 0.90 | 0.89 | 0.90 | 0.91 | 0.90 | 0.88 | 0.90 | 0.90 | 0.90 | 0.88 | 0.90 | 0.90 | 0.90 | 0.88 | 0.90 | 0.90 |
| Boltz | 0.86 | 0.85 | 0.86 | 0.86 | 0.86 | 0.84 | 0.86 | 0.86 | 0.86 | 0.84 | 0.86 | 0.86 | 0.86 | 0.84 | 0.86 | 0.86 |
| Chai | 0.86 | 0.85 | 0.87 | 0.86 | 0.86 | 0.84 | 0.86 | 0.86 | 0.86 | 0.84 | 0.86 | 0.86 | 0.86 | 0.84 | 0.86 | 0.86 |
| OpenFold | 0.64 | 0.62 | 0.64 | 0.63 | 0.64 | 0.62 | 0.64 | 0.62 | 0.64 | 0.62 | 0.64 | 0.62 | 0.64 | 0.62 | 0.64 | 0.62 |
| Proteinx | 0.82 | 0.82 | 0.82 | 0.82 | 0.82 | 0.82 | 0.82 | 0.82 | 0.82 | 0.82 | 0.82 | 0.82 | 0.82 | 0.82 | 0.82 | 0.82 |
| UniFold | 0.57 | 0.39 | 0.67 | 0.49 | 0.56 | 0.38 | 0.66 | 0.48 | 0.56 | 0.38 | 0.66 | 0.48 | 0.56 | 0.38 | 0.66 | 0.48 |

Table 11: Drug discovery performance under different confidence thresholds: actifpTM and PAE.

| PSPM | actifpTM $\geq 0.7$ | | actifpTM $\geq 0.9$ | | PAE $< 15$Å | | PAE $< 10$Å | |
|---|---|---|---|---|---|---|---|---|
| | MAE | $R$ | MAE | $R$ | MAE | $R$ | MAE | $R$ |
| AF2 | 0.173 | 0.979 | 0.170 | 0.982 | 0.168 | 0.980 | 0.163 | 0.983 |
| AF3 | 0.062 | 0.994 | 0.059 | 0.996 | 0.061 | 0.991 | 0.058 | 0.993 |
| Boltz | 0.087 | 0.990 | 0.085 | 0.991 | 0.099 | 0.987 | 0.089 | 0.989 |
| Chai | 0.102 | 0.990 | 0.101 | 0.991 | 0.109 | 0.988 | 0.105 | 0.991 |
| OmegaFold | 0.163 | 0.982 | 0.165 | 0.981 | 0.148 | 0.981 | 0.146 | 0.984 |
| OpenFold | 0.162 | 0.983 | 0.167 | 0.982 | 0.159 | 0.980 | 0.165 | 0.981 |
| Proteinx | 0.163 | 0.984 | 0.168 | 0.983 | 0.162 | 0.982 | 0.167 | 0.981 |
| DeepFold | 0.160 | 0.985 | 0.166 | 0.984 | 0.164 | 0.983 | 0.165 | 0.984 |
| ESMFold | 0.078 | 0.992 | 0.079 | 0.990 | 0.080 | 0.991 | 0.079 | 0.992 |
| RoseTTAFold | 0.200 | 0.974 | 0.207 | 0.973 | 0.198 | 0.972 | 0.201 | 0.974 |
| UniFold | 0.190 | 0.978 | 0.198 | 0.977 | 0.192 | 0.976 | 0.197 | 0.975 |

## 4.5 COMPUTE INFRASTRUCTURE AND RESOURCE UTILIZATION

All experiments in our benchmark were conducted using NVIDIA A100 GPUs. We adopt the default configuration settings provided by each model to ensure consistency and ease of replication. The total

Table 12: Drug discovery performance under combined structural confidence filters: pLDDT≥ 50, actifpTM ≥ 0.9, and PAE<10Å.

| PSPM | pLDDT & actifpTM | | actifpTM & PAE | | pLDDT & PAE | | pLDDT & actifpTM & PAE | |
|---|---|---|---|---|---|---|---|---|
| | MAE | $R$ | MAE | $R$ | MAE | $R$ | MAE | $R$ |
| AF2 | 0.173 | 0.977 | 0.165 | 0.979 | 0.175 | 0.984 | 0.164 | 0.984 |
| AF3 | 0.058 | 0.993 | 0.067 | 0.994 | 0.067 | 0.997 | 0.050 | 0.994 |
| Boltz | 0.087 | 0.993 | 0.090 | 0.989 | 0.085 | 0.990 | 0.087 | 0.993 |
| Chai | 0.106 | 0.992 | 0.104 | 0.992 | 0.117 | 0.990 | 0.104 | 0.990 |
| OmegaFold | 0.146 | 0.980 | 0.162 | 0.983 | 0.157 | 0.982 | 0.169 | 0.981 |
| OpenFold | 0.159 | 0.982 | 0.165 | 0.980 | 0.160 | 0.981 | 0.164 | 0.981 |
| Proteinx | 0.160 | 0.983 | 0.166 | 0.981 | 0.162 | 0.982 | 0.168 | 0.980 |
| DeepFold | 0.159 | 0.984 | 0.161 | 0.983 | 0.164 | 0.982 | 0.165 | 0.983 |
| ESMFold | 0.072 | 0.995 | 0.074 | 0.993 | 0.073 | 0.994 | 0.075 | 0.993 |
| RoseTTAFold | 0.195 | 0.973 | 0.194 | 0.972 | 0.196 | 0.974 | 0.197 | 0.973 |
| UniFold | 0.180 | 0.976 | 0.188 | 0.975 | 0.186 | 0.976 | 0.190 | 0.974 |

computation time across all tasks amounts to approximately 2,000 GPU hours. Detailed instructions, including execution scripts, are provided in our repository to support reproducibility on comparable hardware.

## 5. USER PORTAL FOR STRUCTURE AND TASK EVALUATION

We provide detailed instructions for using the portal to obtain desired protein structures and compare performance across downstream tasks. The portal consists of three primary components: (1) User Input Panel – Includes a navigation panel, a protein ID input field, and a PSPM (Protein Structure Prediction Model) selection interface. (2) Visualization Panel – Contains an Root Mean Square Deviation (RMSD)-based alignment heatmap and a 3D protein structure viewer. (3) Downstream Task Comparison Panel – Displays prediction results from selected PSPM models for relevant biological tasks.

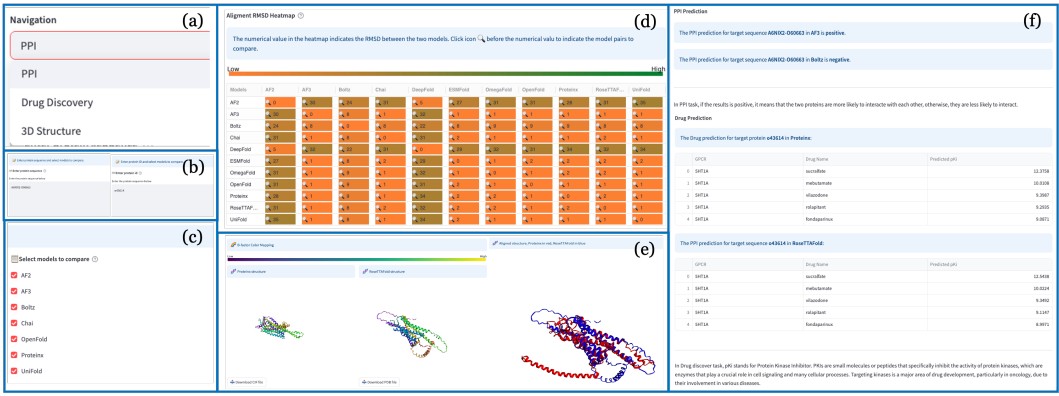

Figure 12: Portal Overview. (a) Navigation for task type. (b) Protein input. (c) Model selection. (d) Structural alignment heatmap. (e) 3D visualization and residue-level uncertainty. (f) Downstream task predictions.

A screenshot of the portal interface is shown in Figure 12. Users can follow the steps below to interact with the portal:

1. **Select a Downstream Task** Users begin by selecting the downstream task of interest. The portal currently supports three options: PPI (Protein-Protein Interaction), Drug Discovery, 3D Structure Access (providing access to a database of 3D protein structures and intrinsically disordered regions, or IDRs)

2. **Input Protein ID and Select PSPMs** Once a task is selected, the protein ID input panel (Figure 8.b) and the PSPM selection panel (Figure 8.e) become available. Users must enter a valid protein ID according to the input format hint and choose at least two PSPMs for comparison.

3. **Visualize RMSD Alignment and Protein Structures** Upon submission, the portal computes an RMSD alignment heatmap, which is displayed in the heatmap panel (Figure 8.d). Users can explore structure comparisons by selecting a pair of PSPMs using the magnifier icon. This action reveals the corresponding 3D protein structures in the visualization panel (Figure 8.e).

4. **View Downstream Task Results** Below the visualization panel, prediction results related to the selected downstream task are displayed:

• For PPI, predictions are shown as binary outcomes (positive or negative).

• For Drug Discovery, results are presented in a tabular format listing the top-ranked drug candidates.

## 6. FUNCTIONAL INSIGHTS FROM DISPROTBENCH PORTAL

To support in-depth, application-driven evaluation of protein structure prediction models, the DisProtBench Portal offers interactive tools that go beyond aggregate metrics. For each model prediction, the portal provides binary PPI outcomes—indicating whether two proteins are predicted to interact—as well as structural similarity assessments via RMSD scores between predicted and reference structures.

These features are particularly valuable in several use cases. First, for drug discovery and target validation, researchers can use the interaction predictions to identify candidate receptor-ligand or protein-protein pairs relevant to disease pathways. Being able to assess whether structural models consistently predict interaction under varying degrees of disorder helps filter out false positives early in the pipeline.

Second, in model selection and error diagnosis, the RMSD-based alignment heatmaps help users identify where structural discrepancies are concentrated, especially within IDRs. This allows developers to pinpoint weaknesses in specific PSPMs, such as unstable modeling of flexible loops or disorder-mediated interfaces, and iterate accordingly.

Finally, for functional hypothesis generation, researchers can visually explore how structural confidence and disorder impact downstream tasks (e.g., loss of binding due to disorder in the interface), providing biologically interpretable insights that static performance scores cannot capture. By allowing configurable, task-specific exploration without requiring local model execution, the portal facilitates both expert and non-expert engagement with the benchmark and supports hypothesis-driven biomedical research.

## 7. USAGE OF LLM

Large language models (LLMs) were employed in a limited and transparent manner during the preparation of this manuscript. Specifically, LLMs were used to assist with linguistic refinement, style adjustments, and minor text editing to improve clarity and readability. They were not involved in formulating the research questions, designing the theoretical framework, conducting experiments, or interpreting results. All scientific contributions—including conceptual development, methodology, analyses, and conclusions—are the sole responsibility of the authors.

