# OpenReview forum: "DISPROTBENCH: A Disorder-Aware, Task-Rich Benchmark for Evaluating Protein Structure Prediction in Realistic Biological Contexts"
_ICLR.cc/2026/Conference — ICLR 2026 Conference Withdrawn Submission_

### Official Review · Reviewer_foFa · 2025-10-27

**Soundness:** 3
**Presentation:** 2
**Contribution:** 3
**Rating:** 4
**Confidence:** 4

**Summary:**

This paper presents a benchmark, referred to as DisProtBench, for evaluating Protein Structure Prediction Models (PSPMs) under conditions of structural disorder and biological complexity. DisProtBench is designed to offer appropriate data complexity, task diversity, and high interpretability. Through empirical evaluation of representative PSPMs on the proposed benchmark, the authors find that global accuracy metrics often fail to reflect task performance in disordered regions.

**Strengths:**

1. The proposed benchmark has significant practical value, as evaluating the performance of PSPMs on intrinsically disordered regions (IDRs) is an important topic in protein modeling.

2. In addition to the datasets, the work provides a visual and interactive interface that facilitates real-world deployment and practical use.

**Weaknesses:**

The main weaknesses of this paper lie in its presentation and clarity:

* The text in **Figure 1** is too small to read comfortably.
* The motivation for including DisProt-annotated disordered regions, GPCR–ligand interactions, and multimeric protein complexes is unclear. The rationale for combining these three categories within a single benchmark should be explained in more details.
* Given that most ICLR readers come from the machine learning community, the paper should provide sufficient explanations or references for domain-specific concepts that fall outside the core ML literature, such as **FEM** (line 200), **HINT**, and **ClinVar** (line 219). At least, corresponding citations should be included.
* In **Figure 4(a)**, there appear to be two instances of “pLDDT ≥ 50.”
* “**Proteinx**” should be corrected to “**Protenix**.”
* In **Figure 6**, should the variable \( R \) actually be \( R^2 \)? Please clarify.
* It is difficult to distinguish the bars corresponding to different models in the figures, as the colors are too similar.
* **Line 404:** The statement “First, confidence-based ... structure.” lacks sufficient justification. Please indicate which figure or table supports this conclusion.

**Questions:**

How is the 95% CI calculated in the experiments?

---

### Official Review · Reviewer_gGnR · 2025-10-31

**Soundness:** 2
**Presentation:** 2
**Contribution:** 2
**Rating:** 2
**Confidence:** 4

**Summary:**

This paper introduces DisProtBench, a benchmark designed for evaluating protein complex structure prediction models from the perspective of intrinsically disordered regions (IDRs) in proteins. The authors curate biologically grounded datasets that emphasize key targets in protein–protein and protein–ligand interactions, particularly those relevant to drug discovery, such as GPCRs and IDR-mediated binding events. The benchmark evaluates a range of state-of-the-art structure prediction models on two tasks: PPI (protein–protein interaction) and Drug Discovery/PLI (protein–ligand interaction). The results show that leading folding models (e.g., AlphaFold 3) achieve higher evaluation scores when structured regions are filtered out in the PPI task, while maintaining robust performance in PLI. In addition, the authors provide an accessible, information-rich web interface for exploring the benchmark results.

**Strengths:**

1. The paper addresses an important problem in biomolecular modeling to understand and evaluate predictions involving IDP regions. Developing benchmarks and metrics that capture IDP-related structural and functional outcomes is a key step toward making current structure prediction models more useful in realistic biological and drug discovery contexts.
2. The benchmark provides a comprehensive coverage of state-of-the-art models in this domain, including both open-source and proprietary methods, enabling a fair and wide-ranging comparison.
3. The authors curate new datasets specifically focused on IDP-related interactions, which could fill a clear gap in existing benchmarks.
4. The inclusion of a user-friendly web interface, if fully realized, would further enhance the benchmark’s accessibility and impact for the broader community.

**Weaknesses:**

1. The manuscript shows several inconsistencies: (1) three curated datasets are mentioned but not clearly linked to the reported results, which instead focus on two tasks; (2) different models are used across experiments without clear explanation of the choice.
2. While the IDP perspective is valuable, the results do not directly address the central questions: what metrics or evaluation presented in this work constitute a meaningful improvement in modeling IDR effects and how can researcher use them as guidances for future model development. In addition, the robustness to pLDDT filtering in the drug-discovery task also seems to contradict the stated hypothesis about the biological role of disorder and flexibility - where more discussion would be appreciated.
3. The main takeaway, that filtering low-confidence (low-pLDDT) regions improves performance, largely aligns with existing evaluation practices (e.g., AlphaFold3 reports values at different confidence levels).
4. Clarity: task and metric definitions (notably Section 5.2 and 5.3) are too coarse to set a clear understanding of the task and evaluation. It should be expanded with more details.

The benchmark is promising and relevant, but the current version feels incomplete and lacks a clear, validated framework for assessing IDP-related predictions. Clarifying datasets, tasks, evaluation metrics and improvements from them would substantially strengthen the paper.

**Questions:**

1. The paper introduces three curated datasets: DisProt-based, individual protein, and disorder-mediated protein-protein interaction sets, and there are some cases where IDPs are functionally important (e.g., agonist/antagonist specificity). However, the results section discusses only two tasks (PPI and ligand binding) without mapping them back to these datasets. How are these tasks related to the proposed datasets and are there any case studies you can provide?
2. The PPI dataset is constructed using AlphaFold-Multimer, which may introduce structural biases toward AF-predicted conformations. Is there any potential impact of this choice?
3. pLDDT is used as a surrogate for flexibility. While the paper shows it correlates well with disorder, it remains a proxy. Given the focus on realistic biological contexts, it would strengthen the work to separately report the group of proteins with experimentally verified disorder labels from those with pLDDT-predicted ones.
4. Focusing on stratifying results by pLDDT thresholds effectively shifts attention to structured regions rather than IDP regions. Although this demonstrates the influence of confidence filtering, such findings are already established (e.g., in AlphaFold3 analyses) and add limited new insight. Can the authors provide more reasoning behind this?
5. The set of benchmarked models appears inconsistent. Section 5.2 lists older models but omits recent ones such as AlphaFold3, while protein language models (ESM-2, ProtBERT, ProtT5) are also mentioned but not shown in results. Please clarify which models were actually evaluated.
6. GPCR-related results are mentioned but not presented. Have the authors specifically looked into this subset?
7. The definitions of the two tasks remain vague. What are the exact prediction objectives (classification vs. regression), and how are interface-level metrics computed?
8. Co-folding models like AlphaFold3, Boltz, Chai, and Protenix can generate ligand conformations, whereas others predict only protein structures. How were these different modeling capabilities handled during evaluation?
9. For the Drug Discovery evaluation pipeline, how are structures encoded in the Evoformer? Does encoding protein and ligand separately overlook key complex-level spatial relationships (e.g., their relative locations)?


[Formats]
- Citations need proper formatting.
- Figure 4a: the x-axis labels contains two "pLDDT ≥ 50" and please specify the actual metric plotted.
- AlphaFold 3 appears in Table 1 but not in Figure 4b.

---

### Official Review · Reviewer_YzHA · 2025-11-01

**Soundness:** 2
**Presentation:** 3
**Contribution:** 2
**Rating:** 4
**Confidence:** 3

**Summary:**

The paper introduces a benchmark to evaluate protein structure prediction models, specifically for intrinsically disordered regions, which lack stable three-dimensional structures in physiological conditions. Since protein structure models generate only one structure for these variable regions, it has been of interest in various studies how to use and interpret structure predictions of these regions. The authors created a dataset of proteins with IDR regions. Then, they utilized predictions from various models (more than 10) for multiple downstream tasks involving protein-protein and protein-ligand interactions. The main finding is that downstream task performance varies between IDR and non-IDR, with lower performance in IDR.

Editorial issues:
  - Figure 4 label on x-axis seems incorrect, repeated mention of pLDDT>50.

**Strengths:**

- The benchmark is based on biology, which is valuable for providing the required domain knowledge to AI tasks. However, the validity of the bioinformatics protocols used to generate the data is difficult to evaluate and is beyond the expertise of AI conference reviewers.
- Evaluation methods are comprehensive, including precision, recall, F, etc.
- The paper presents an entire framework from data processing to the downstream task. Such a framework would be valuable for future work.
- The authors claim to provide code for reproduction and interfaces, which is a valuable contribution to this field.

**Weaknesses:**

- The use of pLDDT is a valuable approach. However, it seems like several other previous works have used a similar approach and it's not clear if it can be considered a novel contribution.
  - There is a lack of discussion on related work in the area of IDR. For example, Ruff, K. M., & Pappu, R. V. (2021). AlphaFold and implications for intrinsically disordered proteins. Journal of molecular biology, 433(20), 167208., Alderson, T. R., Pritišanac, I., Kolarić, Đ., Moses, A. M., & Forman-Kay, J. D. (2023). Systematic identification of conditionally folded intrinsically disordered regions by AlphaFold2. Proceedings of the National Academy of Sciences, 120(44), e2304302120.
  - It would be helpful to have a comparison of the proposed dataset with other datasets, such as Interpro and disport (https://disprot.org), which also include IDR.
  - While the change in performance on the PPI task in Boltz is clear in Fig. 4, there are no significant differences for different pLDDT thresholds in the drug discovery task—as shown in Table 1, Figures 5 and 6. It is unclear whether the conclusion about the importance of the IDR region's impact on performance is valid for the drug discovery condition. As the authors noted, the F scores change by +0.04 in many cases, raising questions about the significance of this result.

**Questions:**

- RR, LR, RP, and LP across PSPMs are only provided for plDDT >50; including data for full and plddt >30 would be helpful.

---

### Official Review · Reviewer_gPrS · 2025-11-02

**Soundness:** 2
**Presentation:** 2
**Contribution:** 2
**Rating:** 4
**Confidence:** 3

**Summary:**

This paper introduces DisProtBench, a benchmark designed to evaluate how well protein structure prediction models handle intrinsically disordered regions (IDRs). The benchmark is curated from DisProt and other datasets, providing residue-level disorder annotations and evaluating multiple structure predictors across disorder-related tasks.

**Strengths:**

* Evaluating disorder is an important and often overlooked challenge in current protein structure predictors. Having a standardized resource is genuinely valuable.

* Multiple mainstream predictors are evaluated, and the discussion touches on model robustness and potential biological implications.

**Weaknesses:**

My biggest concern is around the soundness of using pLDDT as a proxy for structural uncertainty and intrinsic
disorder, specifically:
1. The paper does not specify which dataset Figure 3 is based on. If this figure includes proteins already seen by AlphaFold during training, the strong separation of pLDDT between ordered and disordered residues could be partly explained by data familiarity. It’s unclear if AlphaFold’s high pLDDT values for structured regions still hold for unseen proteins. Clarifying this is important.
2. Are the disorder regions from the predicted structures inferred from pLDDT scores? If so, I guess the confidence from each benchmarked model is used. Are they directly comparable? Why not determine the disorder region based on predicted structures?

**Questions:**

1. Why not compare LDDT or iLDDT, which are standard metrics from structure prediction tasks?

2. Versions or checkpoints of Protenix, Chai, and Boltz are not listed. Since these models have undergone rapid iteration, version numbers or commit hashes are necessary for reproducibility.

3. In figure 4a, the x-axis labels both read “pLDDT > 50.” This seems to be a typo or plotting error.

---

### Note · Authors · 2025-11-12

I have read and agree with the venue's withdrawal policy on behalf of myself and my co-authors.